**Data Availability Statement:** All relevant data are within the manuscript and its Supporting information files.

# COVID-19 and COVID-19 vaccination experiences and perceptions among health workers during the pandemic in Ebonyi state, Nigeria: An analytical cross-sectional study

**Ugwu I. Omale** [1]*, **Cordis O. Ikegwuonu** [1], Glory E. Nkwo[2], Ugochi I. A. Nwali[1], Olaedo O. Nnachi[1], Okechukwu O. Ukpabi[1], Ifeyinwa M. Okeke[1], Richard L. Ewah[3,4], Osarhiemen Iyare[1], Chidinma I. Amuzie[2], Onyinyechukwu U. Oka[1], Victor U. Uduma[5], Azuka S. Adeke[1]

1 Department of Community Medicine, Alex Ekwueme Federal University Teaching Hospital Abakaliki (AEFUTHA), Abakaliki, Ebonyi State, Nigeria, 2 Department of Community Medicine, Federal Medical Centre Umuahia, Umuahia, Abia State, Nigeria, 3 Department of Anaesthesia, Alex Ekwueme Federal University Teaching Hospital Abakaliki (AEFUTHA), Abakaliki, Ebonyi State, Nigeria, 4 Department of Surgery, Anaesthesia Unit, Ebonyi State University, Abakaliki, Ebonyi State, Nigeria, 5 Department of Internal Medicine, Alex Ekwueme Federal University Teaching Hospital Abakaliki (AEFUTHA), Abakaliki, Ebonyi State, Nigeria

* omaleiu@gmail.com

## Abstract

### Background

COVID-19 continues to be a disease of global public health importance and requires long-term management and control. Health workers' (previous) experiences and perceptions regarding the COVID-19 pandemic and COVID-19 vaccination/vaccination process will influence not only their subsequent use of control measures but also public experiences/perceptions. We explored the COVID-19 and COVID-19 vaccination and the vaccination process experiences and perceptions, and their predictors, among the health workers in Ebonyi state, Nigeria.

### Methods

We conducted an online-offline analytical cross-sectional survey between March 12 and May 9, 2022 among all categories of health workers (clinical/non-clinical, public/private) working/living in Ebonyi state who consented to participate and were selected by convenience/snowballing techniques. A structured electronic questionnaire was used to collect data: self-administered via WhatsApp and interviewer-administered via KoBoCollect for participants who did not have WhatsApp. Data was analysed using descriptive statistics and bivariate/multivariate generalized linear models.

### Results

Of the 1276 health workers surveyed: 55.8% had strong COVID-19 experience and perception, 80.7% had good COVID-19 vaccination expectation and perception, and 87.7% had

**Funding:** The authors received no specific funding for this work.

**Competing interests:** The authors have declared that no competing interests exist.

positive COVID-19 vaccination process experience and perception. The most important predictors of the extent and level of COVID-19 and COVID-19 vaccination and the vaccination process experiences and perceptions were level of place of work (primary-secondary/tertiary), level of attitude towards COVID-19 (vaccination), and level of knowledge about COVID-19. Another important predictor was place of work (public/private).

## Conclusions

The evidence indicate the factors that should guide subsequent policy actions in the strategies to enhance the COVID-19 and COVID-19 vaccination and the vaccination process experiences and perceptions of health workers (and their use of control measures) in Ebonyi state, Nigeria, and other similar contexts. It also indicate factors to be considered by future policy actions regarding similar diseases.

## Introduction

COVID-19 related morbidity and mortality is still occurring around the world, more than four years since the disease emerged by end of 2019 and became a pandemic in early 2020 [1,2]. Although it has been declared to no longer be a public health emergency of international concern [3], COVID-19 is still a fatal disease of global public health importance that requires long-term management and control [1]. Millions of new infections or re-infections and thousands of related deaths continue to occur around the world, especially from new variants of the severe acute respiratory syndrome coronavirus 2 (SARS-CoV-2) which have the potential to cause resurgence [1,2]. Over 503000 COVID-19 cases and 10000 related deaths were confirmed globally between the 28-day period of 7 January and 4 June 2024 [2]. However, these statistics are underestimated as the rate of testing and reporting have reduced globally (and some countries no longer test and report) [1,2].

One of the factors sustaining the COVID-19 pandemic was the reduction in the use and observance of COVID-19 control measures (including non-acceptance of COVID-19 vaccination) by the public, including the health workers, and COVID-19 vaccination with the other preventive measures is one of the strategies for the long-term management of the pandemic [1]. Disease risk perception and confidence in the safety and effectiveness of vaccination and the vaccination process/system are factors that influence vaccination acceptance [4]. Health workers are at particular risk of contracting COVID-19 as a result of their regular and close contact with patients and this risk continues to be real because of their key roles in the management of the new COVID-19 infections or re-infections. Thus, the subsequent use of public health control measures, including COVID-19 vaccination, is very crucial for health workers. The acceptance/uptake of COVID-19 vaccination by health workers will be influenced by their (previous) experiences and perceptions regarding the COVID-19 pandemic, COVID-19 vaccination, and the vaccination process [5,6] and the use of other preventive measures will be influenced by their COVID-19 experiences and perceptions [7].

Strong belief that one or other persons have a disease based only on symptoms, irrespective of laboratory diagnoses, is not uncommon in Ebonyi state (and Nigeria/other African countries) where morbidity and mortality from many common diseases (with known causes based on scientific knowledge) are still attributed to superstitious causes (such as spiritual attacks). Such belief was perhaps more striking during the COVID-19 pandemic due to the

unprecedented misinformation, disinformation, and conspiracy theories about COVID-19 and COVID-19 vaccination and could be an important determinant of healthy behaviours. Such belief was also observed among the health workers who are important opinion leaders on health matters and sources of health information for many people. The experiences and perceptions of health workers regarding the COVID-19 pandemic, COVID-19 vaccination, and the vaccination process are crucial, because of their influence on the experiences and perceptions of the general public, and an understanding of these experiences and perceptions, and the determinants, would be useful in the subsequent planning of tailored COVID-19 behaviour change communication strategies.

It was therefore imperative to explore the experiences and perceptions of health workers regarding the COVID-19 pandemic and COVID-19 vaccination and its processes in Ebonyi state. We carried out an extensive online and offline study to evaluate COVID-19 vaccination acceptance and the determinants among the health workers in Ebonyi state, Nigeria [8] and part of the study also explored the COVID-19 and COVID-19 vaccination and the vaccination process experiences and perceptions, and their predictors, among the health workers during the COVID-19 pandemic in the state.

## Methods

### Study design and setting

The study was an analytical cross-sectional survey conducted between March 12 and May 9, 2022 in Ebonyi state. The study protocol is described elsewhere [8]. Ebonyi state is in the south-east geopolitical zone of Nigeria and had a projected population for 2021 of 3,313,229 based on the 2006 national census figure and a growth rate of 2.8%. Christianity is the most practiced religion. The state is divided into three senatorial zones, 13 Local Government Areas (LGAs) and 171 political wards. As of 2020, there were 784 orthodox health care facilities in the state: 566 public health facilities (two tertiary, 13 secondary and 551 primary) and 218 private health facilities (25 secondary and 193 primary) [9]. The two tertiary facilities were the Alex Ekwueme Federal University Teaching Hospital Abakaliki (AEFUTHA) and the National Obstetrics Fistula Centre (NOFIC). The 13 public secondary facilities were the 13 general hospitals in each of the 13 LGAs of the state while the 25 private secondary facilities were private and missionary hospitals.

### Study participants and data collection

The study participants were the health workers in Ebonyi state, including all categories of health workers, both clinical and non-clinical staff in public and private health care sectors. They include primary health care workers (health attendants, community health extension workers, community health officers, nurses and midwives), orderlies, medical laboratory scientists and technologists, patent medicine vendors, pharmacists and pharmacy technicians, medical doctors, dental therapists, physiotherapists, dietician, general admin department staff, personnel department staff, account department staff, public relation officers, security personnel. Eligible health workers were those who were working or living in Ebonyi state and gave verbal consent. Eligible participants were selected by convenience and snowballing sampling techniques.

Data collection was via health workers survey using a structured self-administered and interviewer-administered questionnaire [8]. The sections of the questionnaire include sociodemographic characteristics; COVID-19 experiences and perceptions; basic knowledge of COVID-19; and attitude towards COVID-19 and COVID-19 vaccination. The electronic version of the questionnaire was programmed using the KoBoToolbox software and pre-tested

among health workers who were later exempted from the survey. The design of the questionnaire was informed by published data and expert validation and pre-test were carried out by the study team [8]. More details of the data management and quality control are in the study protocol [8].

To increase acceptance rate, the investigators first made physical/phone contact with many health workers who were available and/or easily accessible and sought their consent. Thereafter, the web link for the self-administered electronic questionnaire was sent to the private WhatsApp pages of those who gave verbal consent and they were asked to, after completing the questionnaire, forward the web link to other eligible health workers they know within the study area. Interviewers also administered the questionnaire with KoBoCollect installed in android devices to health workers who did not have online contact and those living in rural areas with poor or no internet access. 1276 health workers successfully participated in the survey.

## Data management and statistical analyses

The independent factors were sociodemographic characteristics, professional/work-related attributes, main and most trusted sources of information about COVID-19, level of knowledge of COVID-19, and level of attitude towards COVID-19 (vaccination). The basic knowledge of COVID-19 was assessed using 44 knowledge items: each correct response and incorrect responses were respectively scored "1" and "0"; the highest attainable score was 44 and lowest was zero for each participant; scores of ≥75% of 44 were categorised as good knowledge and <75% were poor knowledge. The attitude towards COVID-19 and COVID-19 vaccination was assessed using 16 attitude items: each item had five response options of strongly disagree, disagree, not sure, agree, and strongly agree and scored from "1" to "5" or "5" to "1" as appropriate; the highest attainable score was 80 and the lowest was 16 for each participant; scores of ≥75% of 80 were categorized as good attitude and <75% were poor attitude.

The main outcomes measures were the extent of COVID-19 experience and perception, level of COVID-19 vaccination expectation and perception, and level of COVID-19 vaccination process experience and perception. 5–8 questionnaire items were used to assess the experiences and perceptions of participants about COVID-19 and COVID-19 vaccination and the vaccination process. Each item had five response options and was scored from 0–4. The scores for the 5–8 items related to each outcome was summed for each participant and scores ≥50% of the total versus <50% were respectively considered to be: strong versus not strong COVID-19 experience and perception; good versus poor COVID-19 vaccination expectation and perception; and positive versus negative COVID-19 vaccination process experience and perception. More details are in the study protocol [8].

"Experience and perception" was explored as particular outcomes or variables due to the understanding that "experience" and "perception" are inextricably linked and influence each other. In the context of the COVID-19 pandemic, someone who have witnessed a case of COVID-19 in the neighbourhood might be more likely to perceive or believe that COVID-19 is real and that it is possible to get infected. Conversely, someone who perceive or believe that COVID-19 is real and that it is possible to get infected, will be more likely to experience (observe facts of) COVID-19 cases in the neighbourhood. This means that someone who perceive or believe that COVID-19 is not real and that it is not possible to get infected, will be more likely not to experience (not to observe facts of) COVID-19 cases in the neighbourhood because any case of COVID-19 in the neighbourhood (base on classical symptoms and or laboratory tests) can more easily be interpreted to be other diseases or "spiritual attack".

The other outcomes were the dichotomized positive versus non-positive categories of COVID-19 and COVID-19 vaccination and the vaccination process experiences and perceptions which were assessed with the five-category 5–8 questionnaire items. These outcomes include: fear of getting COVID-19 (very fearful/a little fearful versus not fearful at all/not fearful/not sure), fear of having severe side-effects from COVID-19 vaccination (not fearful at all/not fearful versus very fearful/a little fearful/not sure), etc.

Statistical analyses were done with Stata/SE version 15.1 (Stata Corp, College Station, TX, USA). Data was summarized using frequencies with proportions (expressed as percentages) and median with inter-quartile range as appropriate. Inferential statistics were done using generalized linear models (GLM) and at 2.5% significance level to correct for multiple comparisons. For dichotomous or categorical independent factors, prevalence difference in the outcomes with 97.5% CI and p-values were computed using binomial identity GLM models with robust standard errors. For continuous independent factors, coefficients in the outcomes with 97.5% CI and p-values were computed using the binomial identity GLM models.

All the independent factors were added to the GLM model in the adjusted analyses. For the binomial identity GLM models that failed to achieve convergence, gaussian identity GLM models were used instead [10].

### Ethics statement

Ethical approval for this study was obtained from the Ebonyi State Health Research and Ethics Committee (EBSHREC/15/01/2022-02/01/2023) and Research and Ethics Committee of Alex Ekwueme Federal University Teaching Hospital Abakaliki (14/12/2021-17/02/2022). Verbal informed consent was obtained from the study participants during which the purpose the study, kind of participation, likely duration of participation, voluntary nature of participation, absence of potential harm, potential benefit, and confidential nature of the study were duly communicated to them.

## Results

### Sociodemographic and background characteristics

The sociodemographic and background (work-related) characteristics of the 1276 health workers who participated in the study are presented in Table 1. Of the 1276 participants, the median age (IQR) was 33 years (26–43) and the median years of working experience was 5 years (2–13). Majority of them were females (67.2%), were married (54.2%), had a tertiary education (56.9%), were clinical staff (87.0%), were working primarily in private health facilities (51.1%), and were working at primary health facilities (74.6%).

### COVID-19 and COVID-19 vaccination and the vaccination process experiences and perceptions

The COVID-19 and COVID-19 vaccination and the vaccination process experiences and perceptions of the 1276 health workers who participated in the study are presented in Table 2. Regarding COVID-19 experiences and perceptions: more of the participants (44.3%) were very fearful about getting COVID-19 followed by those who were a little fearful (19.1%); majority of them (52.7%) had the perception that it was highly possible for them to get COVID-19 followed by those who had the perception that it was not possible at all (19.0%); most of them (81.4%) were sure they had never gotten COVID-19 followed by those who were not sure about it (10.0%); and most of them (86.0%) did not know any person who had gotten COVID-19; etc (Table 2).

**Table 1. Sociodemographic and background characteristics of the 1276 study participants.**

|  | n | % |
|---|---|---|
| Gender |  |  |
| Male | 419 | 32.8 |
| Female | 857 | 67.2 |
| Age, median (IQR), years | 33 (26–43) | – |
| Marital status |  |  |
| Married | 691 | 54.2 |
| Not married[1] | 585 | 45.8 |
| Educational level |  |  |
| No formal education | 10 | 0.8 |
| Primary | 36 | 2.8 |
| Secondary | 504 | 39.5 |
| Tertiary | 726 | 56.9 |
| Work category or cadre |  |  |
| Non-clinical staff[2] | 166 | 13.0 |
| Clinical staff[3] | 1110 | 87.0 |
| Working experience, median (IQR), years | 5 (2–13) | – |
| Primary place of work |  |  |
| Private health facility[4] | 652 | 51.1 |
| Public health facility[5] | 624 | 48.9 |
| Level of primary place of work |  |  |
| Primary health facility[6] | 952 | 74.6 |
| Secondary health facility[7] | 39 | 3.1 |
| Tertiary health facility[8] | 285 | 22.3 |

[1]Separated or Divorced or Widowed or Never married (Single).

[2]Admin, Personnel, Account, Public relation officer, Security etc

[3]Patent medicine vendor, Primary health care worker (Health attendant, Community health extension worker, Community health officer, Nurse & midwife), Orderly, Medical laboratory scientist or technologist, Pharmacist or pharmacy technician, Medical doctor, and others (Dental therapist, physiotherapist, Dietician etc).

[4]Patent medicine vendor (PMV), Private pharmacy, Private laboratory, Private hospital or clinic, Missionary hospital.

[5]Primary health care (PHC) centre, General hospital, Federal tertiary health centre, and Federal university teaching hospital.

[6]PMV, Private pharmacy, Private laboratory, Private hospital or clinic, and PHC centre.

[7]Missionary hospital and General hospital.

[8]Federal tertiary health centre and Federal university teaching hospital.

Regarding COVID-19 vaccination expectations and perceptions: majority of the participants (52.5%) had the perception that it was very important for them to receive COVID-19 vaccination followed by those who had the perception that it was important (24.8%); more of them (33.8%) were not fearful at all of severe side-effects from COVID-19 vaccination followed by those who were very fearful (29.8%); majority of them (58.2%) believed COVID-19 vaccination would give them full protection against COVID-19 followed by those who were not sure about it (16.8%); etc (Table 2). Regarding COVID-19 vaccination process experiences and perceptions: most of the participants (96.0%) said they had heard many times that COVID-19 vaccination was available for them to go and receive followed by those who said they had heard about it few times or once (3.6%); majority of them (56.8%) said they knew a very close

**Table 2. COVID-19 and COVID-19 vaccination and the vaccination process experiences and perceptions among the 1276 study participants.**

| | n | % | | n | % |
|---|---|---|---|---|---|
| **COVID-19 experiences & perceptions** | | | **COVID-19 vaccination expectations & perceptions** | | |
| How fearful are you about getting COVID-19? | | | How important is it for you to receive COVID-19 vaccination? | | |
| Very fearful | 565 | 44.3 | Very important | 670 | 52.5 |
| A little fearful | 244 | 19.1 | Important | 316 | 24.8 |
| Not sure | 51 | 4.0 | Not sure | 124 | 9.7 |
| Not fearful | 227 | 17.8 | Not important | 65 | 5.1 |
| Not fearful at all | 189 | 14.8 | Not important at all | 101 | 7.9 |
| How possible is it for you to get COVID-19? | | | How fearful are you about severe side-effects from the vaccination? | | |
| Highly possible | 673 | 52.7 | Not fearful at all | 431 | 33.8 |
| A bit possible | 150 | 11.8 | Not fearful | 172 | 13.5 |
| Not sure | 54 | 4.2 | Not sure | 106 | 8.3 |
| Not possible | 156 | 12.2 | A little fearful | 187 | 14.6 |
| Not possible at all | 243 | 19.0 | Very fearful | 380 | 29.8 |
| How possible is it for you to get severe COVID-19? | | | What protection against COVID-19 will the vaccination give? | | |
| Highly possible | 545 | 42.7 | Full protection | 742 | 58.2 |
| A bit possible | 154 | 12.1 | Partial protection | 188 | 14.7 |
| Not sure | 82 | 6.4 | Not sure | 214 | 16.8 |
| Not possible | 190 | 14.9 | No protection | 46 | 3.6 |
| Not possible at all | 305 | 23.9 | No protection at all | 86 | 6.7 |
| Have you ever had COVID-19? | | | How do you trust the health workers giving the vaccination? | | |
| Yes, surely | 38 | 3.0 | Trust them very much | 694 | 54.4 |
| Yes, think so | 21 | 1.6 | Trust them | 316 | 24.7 |
| Not sure | 128 | 10.0 | Not sure | 121 | 9.5 |
| No, think so | 51 | 4.0 | Do not trust them | 51 | 4.0 |
| No, surely | 1038 | 81.4 | Do not trust them at all | 94 | 7.4 |
| Have you ever had severe COVID-19? | | | How do you trust the government providing the vaccination? | | |
| Yes, very serious | 8 | 0.6 | Trust them very much | 622 | 48.7 |
| Yes, a bit serious | 12 | 0.9 | Trust them | 297 | 23.3 |
| Not sure | 13 | 1.0 | Not sure | 122 | 9.6 |
| No, not serious | 16 | 1.3 | Do not trust them | 103 | 8.1 |
| No, not serious at all | 10 | 0.8 | Do not trust them at all | 132 | 10.3 |
| | | | **COVID-19 vaccination process experiences & perceptions** | | |
| Know any person who have had COVID-19? | | | Ever heard COVID-19 vaccination was available for receipt? | | |
| Yes, a very close person | 45 | 3.5 | Yes, many times | 1225 | 96.0 |
| Yes, a close person | 49 | 3.8 | Yes, once/few times | 46 | 3.6 |
| Yes, a distant person | 41 | 3.2 | Not sure | 4 | 0.3 |
| Yes, a very distant person | 45 | 3.5 | No, no time | 1 | 0.1 |
| No | 1096 | 86.0 | No, no time at all | 0 | 0 |
| Know any person who have had severe COVID-19? | | | Know a COVID-19 vaccination place? | | |
| Yes, a very close person | 31 | 2.4 | Yes, a very close place | 725 | 56.8 |
| Yes, a close person | 30 | 2.4 | Yes, a close place | 202 | 15.8 |
| Yes, a distant person | 30 | 2.4 | Yes, a far place | 171 | 13.4 |
| Yes, a very distant person | 43 | 3.4 | Yes, a very far place | 50 | 4.0 |
| No | 46 | 3.6 | No | 128 | 10.0 |
| Know any person who have died from COVID-19? | | | Frequency of COVID-19 vaccination at that place? | | |
| Yes, a very close person | 17 | 1.3 | Daily, down to twice a week | 725 | 56.8 |
| Yes, a close person | 13 | 1.0 | Once a week | 22 | 1.7 |

*(Continued)*

**Table 2.** (Continued)

| | n | % | | n | % |
|---|---|---|---|---|---|
| Yes, a distant person | 35 | 2.7 | Once in two–four weeks | 12 | 0.9 |
| Yes, a very distant person | 45 | 3.5 | No fixed time | 30 | 2.4 |
| No | 70 | 5.5 | Do not know | 359 | 28.1 |
| | | | Queue at the vaccination place? | | |
| | | | No queue | 631 | 49.4 |
| | | | Short queue | 220 | 17.2 |
| | | | Do not know | 270 | 21.2 |
| | | | Long queue | 25 | 2.0 |
| | | | Very long queue | 2 | 0.2 |
| | | | How caring are the health workers at the vaccination place? | | |
| | | | Very caring | 615 | 48.2 |
| | | | Caring | 269 | 21.1 |
| | | | Not sure | 260 | 20.4 |
| | | | Not caring | 1 | 0.1 |
| | | | Not caring at all | 3 | 0.2 |

COVID-19 vaccination place/site followed by those who said they knew a close place (15.8%); etc (Table 2).

## Predictors of COVID-19 and COVID-19 vaccination and the vaccination process experiences and perceptions

Prevalence estimates and crude and adjusted prevalence differences (for categorical independent factors) and coefficients (for continuous independent factors) and their respective 97.5% CI and p-values are presented in Tables 3–5. The crude and adjusted p-values of the overall effects of polychotomous independent factors are also presented.

The extent of COVID-19 experience and perception and the associations between it and sociodemographic and background factors are presented in Table 3. Among the 1276 study participants, 55.8% had strong COVID-19 experience and perception while 44.2% had not strong COVID-19 experience and perception. As shown by the adjusted results, the predictors of strong COVID-19 experience and perception were: being a health worker at a primary/secondary health facility (adjusted prevalence difference (aPD) 53.8%, 97.5% CI 46.0–61.7, p<0.0001); being a non-clinical health worker (aPD 10.7%, 2.9–18.4, p = 0.0022); good attitude towards COVID-19 (vaccination) (aPD 47.0%, 40.7–53.2, p<0.0001); good knowledge about COVID-19 (aPD 13.0%, 7.4–18.6, p<0.0001); and age as one year increase in age increases the probability of having strong COVID-19 experience and perception by 0.4% (adjusted coefficient (aCoef) 0.4%, 97.5% CI 0.01–0.9), p = 0.0218).

The level of COVID-19 vaccination expectation and perception and the associations between it and sociodemographic and background factors are presented in Table 4. Among the 1276 study participants, 80.7% had good COVID-19 vaccination expectation and perception while 19.3% had poor COVID-19 vaccination expectation and perception. The predictors of good COVID-19 vaccination expectation and perception were: being a health worker at a primary/secondary health facility (aPD 27.3%, 19.8–34.8, p<0.0001); being a health worker at a public health facility (aPD 6.0%, 1.7–10.3, p = 0.0017); good attitude towards COVID-19 (vaccination) (aPD 49.2%, 41.6–56.8, p<0.0001); and good knowledge about COVID-19 (aPD 7.4%, 3.1–11.7, p = 0.0001).

**Table 3. Association between sociodemographic and background factors and the extent of COVID-19 experience and perception among the 1276 study participants.**

| | Extent of COVID-19 experience & perception* | | Crude results | | Adjusted results** | |
|---|---|---|---|---|---|---|
| | Strong n (%) 712 (55.8) | Not strong n (%) 564 (44.2) | cPD (97.5% CI) or cCoef (97.5% CI) | p value | aPD (97.5% CI) or aCoef (97.5% CI) | p value |
| Gender | | | | | | |
| Female | 490 (57.2) | 367 (42.8) | 0 | – | 0 | – |
| Male | 222 (53.0) | 197 (47.0) | -4.2% (-10.8–2.5) | 0.1577 | 1.4% (-4.0–6.8) | 0.5687 |
| Age, years (coefficient) | – | – | 0.2% (-0.1–0.5) | 0.1372 | 0.4% (0.01–0.9) | 0.0218 |
| Marital status | | | | | | |
| Not married[1] | 319 (54.5) | 266 (45.5) | 0 | – | 0 | – |
| Married | 393 (56.9) | 298 (43.1) | 2.3% (-3.9–8.6) | 0.4011 | 1.4% (-5.0–7.8) | 0.6298 |
| Educational level | | | | | | |
| None, primary, or secondary | 335 (60.9) | 215 (39.1) | 0 | – | 0 | – |
| Tertiary | 377 (51.9) | 349 (48.1) | -9.0% (-15.2–(-2.7)) | 0.0013 | 3.0% (-3.4–9.4) | 0.2960 |
| Work category | | | | | | |
| Clinical staff | 620 (55.9) | 490 (44.1) | 0 | – | 0 | – |
| Non-clinical staff | 92 (55.4) | 74 (44.6) | -0.4% (-9.7–8.8) | 0.9164 | 10.7% (2.9–18.4) | 0.0022 |
| Working experience, years (coefficient) | – | – | 0.2% (-0.2–0.6) | 0.2485 | -0.5% (-1–0.04) | 0.0376 |
| Primary place of work | | | | | | |
| Private health facility[2] | 420 (64.4) | 232 (35.6) | 0 | – | | |
| Public health facility[3] | 292 (46.8) | 332 (53.2) | -17.6% (-23.8–(-11.5)) | <0.0001 | 0.6% (-5.9–7.2) | 0.8366 |
| Level of primary place of work | | | | | | |
| Tertiary health facility[4] | 42 (14.7) | 243 (85.3) | 0 | – | 0 | – |
| Primary health facility[5] or secondary health facility[6] | 670 (67.6) | 321 (32.4) | 52.9% (47.1–58.6) | <0.0001 | 53.8% (46.0–61.7) | <0.0001 |
| Main source of information about COVID-19 | | | | 0.0033$ | | 0.9023$ |
| Internet, social media (whatsapp, facebook), & SMS | 70 (43.5) | 91 (56.5) | 0 | – | 0 | – |
| Traditional media (television, radio, prints) | 291 (58.0) | 211 (42.0) | 14.5% (4.4–24.5) | 0.0012 | 0.7% (-8.1–9.6) | 0.8523 |
| Interpersonal[7] | 351 (57.3) | 262 (42.7) | 13.8% (3.9–23.6) | 0.0017 | -0.8% (10.6–9.0) | 0.8484 |
| Most trusted source of information about COVID-19 | | | | 0.0009$ | | 0.7794$ |
| Internet, social media (whatsapp, facebook), & SMS | 54 (44.3) | 68 (55.7) | 0 | – | 0 | – |
| Traditional media (television, radio, prints) | 317 (61.2) | 201 (38.8) | 16.9% (5.8–28.1) | 0.0007 | 2.8% (-6.5–12.2) | 0.4973 |
| Interpersonal[7] | 341 (53.6) | 295 (46.4) | 9.4% (-1.7–20.4) | 0.0570 | 2.8% (-7.4–12.9) | 0.5429 |
| Level of knowledge about COVID-19[8] | | | | | | |
| Poor | 375 (50.9) | 361 (49.1) | 0 | – | 0 | – |
| Good | 337 (62.4) | 203 (37.6) | 11.5% (5.2–17.7) | <0.0001 | 13.0% (7.4–18.6) | <0.0001 |
| Level of attitude towards COVID-19 (vaccination)[9] | | | | | | |
| Poor | 21 (8.8) | 218 (91.2) | 0 | – | 0 | – |

*(Continued)*

**Table 3.** (Continued)

| | Extent of COVID-19 experience & perception* | | Crude results | | Adjusted results** | |
|---|---|---|---|---|---|---|
| | Strong n (%) 712 (55.8) | Not strong n (%) 564 (44.2) | cPD (97.5% CI) or cCoef (97.5% CI) | p value | aPD (97.5% CI) or aCoef (97.5% CI) | p value |
| Good | 691 (66.6) | 346 (33.4) | 57.8% (52.6–63.1) | <0.0001 | 47.0% (40.7–53.2) | <0.0001 |

cPD = Crude prevalence difference. aPD = Adjusted prevalence difference. cCoef = Crude coefficient. aCoef = Adjusted coefficient.

*COVID-19 experiences and perceptions score of ≥50% of the highest attainable score of 32 was strong experience and perception and <50% was not strong experience and perception.

**Adjusted for Basic knowledge of COVID-19; Attitude towards COVID-19 & COVID-19 vaccination; Source of information about COVID-19 (Main source and Most trusted source of information about COVID-19); Sociodemographic characteristics (Gender, Age, Marital status, Educational level); and Work related attributes (Work category, Years of working experience, Primary place of work (public and private), Level of primary place of work (primary, secondary, tertiary).

$p value of overall effect.

[1]Separated or Divorced or Widowed or Never married (Single).

[2]Patent medicine vendor (PMV), Private pharmacy, Private laboratory, Private hospital or clinic, Missionary hospital.

[3]Primary health care (PHC) centre, General hospital, Federal tertiary health centre, and Federal university teaching hospital.

[4]Federal tertiary health centre and Federal university teaching hospital.

[5]PMV, Private pharmacy, Private laboratory, Private hospital or clinic, and PHC centre.

[6]Missionary hospital and General hospital.

[7]Relatives/friends, health workers, place of work, place of worship etc.

[8]Knowledge score of <75% of the highest attainable score of 44 was poor knowledge and ≥75% was good knowledge.

[9]Attitude score of <75% of the highest attainable score of 80 was poor attitude and ≥75% was good attitude.

The level of COVID-19 vaccination process experience and perception and the associations between it and sociodemographic and background factors are presented in Table 5. Among the 1276 study participants, 87.7% had positive COVID-19 vaccination process experience and perception while 12.3% had negative COVID-19 vaccination process experience and perception. The predictors of positive COVID-19 vaccination process experience and perception were: being a health worker at a primary/secondary health facility (aPD 9.3%, 3.6–15.0, p = 0.0003); being a health worker at a public health facility (aPD 5.7%, 1.6–9.9, p = 0.0020); having a tertiary education (aPD 5.7%, 0.9–10.5, p = 0.0076); good attitude towards COVID-19 (vaccination) (aPD 21.3%, 14.3–28.4, p<0.0001); and good knowledge about COVID-19 (aPD 11.7%, 7.7–15.6, p<0.0001).

## Predictors of dichotomized (positive and non-positive) COVID-19 and COVID-19 vaccination and the vaccination process experiences and perceptions

These results for the 1276 study participants are presented in S1 Appendix. Regarding COVID-19 experiences and perceptions, 63.4% were fearful of getting COVID-19 while 36.6% were not fearful/not sure and the predictors of being fearful of getting COVID-19 were being a health worker at a primary/secondary health facility, being a health worker at a public health facility, having a tertiary education, good attitude towards COVID-19 (vaccination), poor knowledge of COVID-19, and the most trusted source of information about COVID-19 (S1 Appendix p 2). 64.5% said it was possible for them to get COVID-19 while 35.5% said it was not possible or that they were not sure about it and the predictors of having the perception that it was possible to get COVID-19 were having a tertiary education, good attitude towards

**Table 4. Association between sociodemographic and background factors and the COVID-19 vaccination expectation and perception level among the 1276 study participants.**

| | COVID-19 vaccination expectation & perception level* | | Crude results | | Adjusted results** | |
|---|---|---|---|---|---|---|
| | Good n (%) 1030 (80.7) | Poor n (%) 246 (19.3) | cPD (97.5% CI) or cCoef (97.5% CI) | p value | aPD (97.5% CI) or aCoef (97.5% CI) | p value |
| Gender | | | | | | |
| Female | 698 (81.5) | 159 (18.5) | 0 | – | 0 | – |
| Male | 332 (79.2) | 87 (20.8) | -2.2% (-7.6–3.1) | 0.3542 | 2.8% (-1.7–7.3) | 0.1688 |
| Age, years (coefficient) | – | – | -0.04% (-0.3–0.2) | 0.7368 | -0.1% (-0.5–0.2) | 0.3638 |
| Marital status | | | | | | |
| Not married[1] | 462 (79.0) | 123 (21.0) | 0 | – | 0 | – |
| Married | 568 (82.2) | 123 (17.8) | 3.2% (-1.8–8.2) | 0.1475 | 3.7% (-1.7–9.1) | 0.1278 |
| Educational level | | | | | | |
| None, primary, or secondary | 456 (82.9) | 94 (17.1) | 0 | – | 0 | – |
| Tertiary | 574 (79.1) | 152 (20.9) | -3.8% (-8.8–1.1) | 0.0811 | -1.1% (-5.8–3.6) | 0.6080 |
| Work category | | | | | | |
| Clinical staff | 901 (81.2) | 209 (18.8) | 0 | – | 0 | – |
| Non-clinical staff | 129 (77.7) | 37 (22.3) | -3.5% (-11.2–4.2) | 0.3142 | 0.6% (-6.6–7.9) | 0.8417 |
| Working experience, years (coefficient) | – | – | 0.1% (-0.2–0.4) | 0.3862 | -0.1% (-0.5–0.3) | 0.5110 |
| Primary place of work | | | | | | |
| Private health facility[2] | 541 (83.0) | 111 (17.0) | 0 | – | 0 | – |
| Public health facility[3] | 489 (78.4) | 135 (21.6) | -4.6% (-9.6–0.3) | 0.0370 | 6.0% (1.7–10.3) | 0.0017 |
| Level of primary place of work | | | | | | |
| Tertiary health facility[4] | 168 (59.0) | 117 (41.0) | 0 | – | 0 | – |
| Primary health facility[5] or secondary health facility[6] | 862 (87.0) | 129 (13.0) | 28.0% (21.1–35.0) | <0.0001 | 27.3% (19.8–34.8) | <0.0001 |
| Main source of information about COVID-19 | | | | 0.0157$ | | 0.8582$ |
| Internet, social media (whatsapp, facebook), & SMS | 115 (71.4) | 46 (28.6) | 0 | – | 0 | – |
| Traditional media (television, radio, prints) | 409 (81.5) | 93 (18.5) | 10.1% (1.2–18.9) | 0.0112 | 0.5% (-8.2–9.3) | 0.8920 |
| Interpersonal[7] | 506 (82.5) | 107 (17.5) | 11.1% (2.4–19.8) | 0.0041 | -1.2% (-11.1–8.8) | 0.7925 |
| Most trusted source of information about COVID-19 | | | | 0.0083$ | | 0.1612$ |
| Internet, social media (whatsapp, facebook), & SMS | 85 (69.7) | 37 (30.3) | 0 | – | 0 | – |
| Traditional media (television, radio, prints) | 432 (83.4) | 86 (16.6) | 13.7% (3.7–23.8) | 0.0022 | 7.2% (-2.9–17.3) | 0.1082 |
| Interpersonal[7] | 513 (80.7) | 123 (19.3) | 11.0% (1.0–21.0) | 0.0135 | 9.8% (-1.7–21.2) | 0.0564 |
| Level of knowledge about COVID-19[8] | | | | | | |
| Poor | 560 (76.1) | 176 (23.9) | 0 | – | 0 | – |
| Good | 470 (87.0) | 70 (13.0) | 10.9% (6.2–15.7) | <0.0001 | 7.4% (3.1–11.7) | 0.0001 |
| Level of attitude towards COVID-19 (vaccination)[9] | | | | | | |
| Poor | 86 (36.0) | 153 (64.0) | 0 | – | 0 | – |

(*Continued*)

**Table 4.** (Continued)

| | COVID-19 vaccination expectation & perception level* | | Crude results | | Adjusted results** | |
|---|---|---|---|---|---|---|
| | Good n (%) 1030 (80.7) | Poor n (%) 246 (19.3) | cPD (97.5% CI) or cCoef (97.5% CI) | p value | aPD (97.5% CI) or aCoef (97.5% CI) | p value |
| Good | 944 (91.0) | 93 (9.0) | 55.0% (47.8–62.3) | <0.0001 | 49.2% (41.6–56.8) | <0.0001 |

cPD = Crude prevalence difference. aPD = Adjusted prevalence difference. cCoef = Crude coefficient. aCoef = Adjusted coefficient.

*COVID-19 vaccination expectations and perceptions score of ≥50% of the highest attainable score of 20 was good expectation and perception and <50% was poor expectation and perception.

**Adjusted for Basic knowledge of COVID-19; Attitude towards COVID-19 & COVID-19 vaccination; Source of information about COVID-19 (Main source and Most trusted source of information about COVID-19); Sociodemographic characteristics (Gender, Age, Marital status, Educational level); and Work related attributes (Work category, Years of working experience, Primary place of work (public and private), Level of primary place of work (primary, secondary, tertiary).

$p value of overall effect.

[1]Separated or Divorced or Widowed or Never married (Single).

[2]Patent medicine vendor (PMV), Private pharmacy, Private laboratory, Private hospital or clinic, Missionary hospital.

[3]Primary health care (PHC) centre, General hospital, Federal tertiary health centre, and Federal university teaching hospital.

[4]Federal tertiary health centre and Federal university teaching hospital.

[5]PMV, Private pharmacy, Private laboratory, Private hospital or clinic, and PHC centre.

[6]Missionary hospital and General hospital.

[7]Relatives/friends, health workers, place of work, place of worship etc.

[8]Knowledge score of <75% of the highest attainable score of 44 was poor knowledge and ≥75% was good knowledge.

[9]Attitude score of <75% of the highest attainable score of 80 was poor attitude and ≥75% was good attitude.

COVID-19 (vaccination), good knowledge about COVID-19, the most trusted source of information about COVID-19, and increase in age (S1 Appendix p 4).

Regarding COVID-19 vaccination expectations and perceptions, 77.3% said it was important for them to receive COVID-19 vaccination while 22.7% said it was not important or that they were not sure about it and the predictors of having the perception that it was important to receive COVID-19 vaccination were being a health worker at a primary/secondary health facility, being a health worker at a public health facility, being married, and good attitude towards COVID-19 (vaccination) (S1 Appendix p 6). 47.3% were not fearful of having severe side-effects from COVID-19 vaccination while 52.7% were fearful or not sure about it and the predictors of not being fearful of having severe side-effects from COVID-19 vaccination were being a health worker at a primary/secondary health facility, being a non-clinical health worker, having a tertiary education, good attitude towards COVID-19 (vaccination), good knowledge about COVID-19, the main source of information about COVID-19, and increase in age (S1 Appendix p 8). 72.9% said COVID-19 vaccination would give them protection against COVID-19 while 27.1% said it would give no protection or that they were not sure about it and the predictors of having the perception that COVID-19 vaccination would give protection against COVID-19 were being a health worker at a primary/secondary health facility, having a tertiary education, good attitude towards COVID-19 (vaccination), and the main source of information about COVID-19 (S1 Appendix p 10).

Regarding COVID-19 vaccination process experiences and perceptions, 72.7% knew a close COVID-19 vaccination place while 27.3% knew a far place or no place and the predictors of knowing a close COVID-19 vaccination place were being a health worker at a public health

**Table 5. Association between sociodemographic and background factors and the COVID-19 vaccination process experience and perception level among the 1276 study participants.**

| | COVID-19 vaccination process experience & perception level* | | Crude results | | Adjusted results** | |
|---|---|---|---|---|---|---|
| | Positive n (%) 1119 (87.7) | Negative n (%) 157 (12.3) | cPD (97.5% CI) or cCoef (97.5% CI) | p value | aPD (97.5% CI) or aCoef (97.5% CI) | p value |
| Gender | | | | | | |
| Female | 753 (87.9) | 104 (12.1) | 0 | – | 0 | – |
| Male | 366 (87.4) | 53 (12.6) | -0.5% (-4.9–3.9) | 0.7943 | 0.2% (-4.2–4.6) | 0.9276 |
| Age, years (coefficient) | – | – | 0.2% (0.1–0.4) | 0.0003 | 0.1% (-0.2–0.4) | 0.4445 |
| Marital status | | | | | | |
| Not married[1] | 488 (83.4) | 97 (16.6) | 0 | – | 0 | – |
| Married | 631 (91.3) | 60 (8.7) | 7.9% (3.7–12.1) | <0.0001 | 1.5% (-3.3–6.4) | 0.4792 |
| Educational level | | | | | | |
| None, primary, or secondary | 456 (82.9) | 94 (17.1) | 0 | – | 0 | – |
| Tertiary | 663 (91.3) | 63 (8.7) | 8.4% (4.1–12.7) | <0.0001 | 5.7% (0.9–10.5) | 0.0076 |
| Work category | | | | | | |
| Clinical staff | 973 (87.7) | 137 (12.3) | 0 | – | 0 | – |
| Non-clinical staff | 146 (87.9) | 20 (12.1) | 0.3% (-5.8–6.4) | 0.9137 | 3.8% (-2.6–10.2) | 0.1819 |
| Working experience, years (coefficient) | – | – | 0.5% (0.2–0.7) | <0.0001 | 0.06% (-0.2–0.4) | 0.6535 |
| Primary place of work | | | | | | |
| Private health facility[2] | 548 (84.1) | 104 (15.9) | 0 | – | 0 | – |
| Public health facility[3] | 571 (91.5) | 53 (8.5) | 7.5% (3.4–11.5) | <0.0001 | 5.7% (1.6–9.9) | 0.0020 |
| Level of primary place of work | | | | | | |
| Tertiary health facility[4] | 243 (85.3) | 42 (14.7) | 0 | – | 0 | – |
| Primary health facility[5] or secondary health facility[6] | 876 (88.4) | 115 (11.6) | 3.1% (-2.1–8.4) | 0.1796 | 9.3% (3.6–15.0) | 0.0003 |
| Main source of information about COVID-19 | | | | 0.5577$ | | 0.0390$ |
| Internet, social media (whatsapp, facebook), & SMS | 142 (88.2) | 19 (11.8) | 0 | – | 0 | – |
| Traditional media (television, radio, prints) | 434 (86.5) | 68 (13.5) | -1.7% (-8.4–4.9) | 0.5566 | -6.8% (-14.7–1.1) | 0.0533 |
| Interpersonal[7] | 543 (88.6) | 70 (11.4) | 0.4% (-6.0–6.8) | 0.8934 | -1.6% (-10.6–7.5) | 0.6936 |
| Most trusted source of information about COVID-19 | | | | 0.7501$ | | 0.0670$ |
| Internet, social media (whatsapp, facebook), & SMS | 105 (86.1) | 17 (13.9) | 0 | – | 0 | – |
| Traditional media (television, radio, prints) | 458 (88.4) | 60 (11.6) | 2.4% (-5.1–10.1) | 0.4939 | 7.5% (-1.3–16.2) | 0.0571 |
| Interpersonal[7] | 556 (87.4) | 80 (12.6) | 1.4% (-6.3–9.0) | 0.6902 | 3.4% (-6.7–13.5) | 0.4528 |
| Level of knowledge about COVID-19[8] | | | | | | |
| Poor | 600 (81.5) | 136 (18.5) | 0 | – | 0 | – |
| Good | 519 (96.1) | 21 (3.9) | 14.6% (10.9–18.3) | <0.0001 | 11.7% (7.7–15.6) | <0.0001 |
| Level of attitude towards COVID-19 (vaccination)[9] | | | | | | |
| Poor | 158 (66.1) | 81 (33.9) | 0 | – | 0 | – |

(*Continued*)

**Table 5.** (Continued)

| | COVID-19 vaccination process experience & perception level* | | Crude results | | Adjusted results** | |
|---|---|---|---|---|---|---|
| | Positive n (%) 1119 (87.7) | Negative n (%) 157 (12.3) | cPD (97.5% CI) or cCoef (97.5% CI) | p value | aPD (97.5% CI) or aCoef (97.5% CI) | p value |
| Good | 961 (92.7) | 76 (7.3) | 26.6% (19.5–33.7) | <0.0001 | 21.3% (14.3–28.4) | <0.0001 |

cPD = Crude prevalence difference. aPD = Adjusted prevalence difference. cCoef = Crude coefficient. aCoef = Adjusted coefficient.

*COVID-19 vaccination process experiences and perceptions score of ≥50% of the highest attainable score of 20 was positive experience and perception and <50% was negative experience and perception.

**Adjusted for Basic knowledge of COVID-19; Attitude towards COVID-19 & COVID-19 vaccination; Source of information about COVID-19 (Main source and Most trusted source of information about COVID-19); Sociodemographic characteristics (Gender, Age, Marital status, Educational level); and Work related attributes (Work category, Years of working experience, Primary place of work (public and private), Level of primary place of work (primary, secondary, tertiary).

$^\$$p value of overall effect.

[1]Separated or Divorced or Widowed or Never married (Single).

[2]Patent medicine vendor (PMV), Private pharmacy, Private laboratory, Private hospital or clinic, Missionary hospital.

[3]Primary health care (PHC) centre, General hospital, Federal tertiary health centre, and Federal university teaching hospital.

[4]Federal tertiary health centre and Federal university teaching hospital.

[5]PMV, Private pharmacy, Private laboratory, Private hospital or clinic, and PHC centre.

[6]Missionary hospital and General hospital.

[7]Relatives/friends, health workers, place of work, place of worship etc.

[8]Knowledge score of <75% of the highest attainable score of 44 was poor knowledge and ≥75% was good knowledge.

[9]Attitude score of <75% of the highest attainable score of 80 was poor attitude and ≥75% was good attitude.

facility, good attitude towards COVID-19 (vaccination), and good knowledge about COVID-19 (S1 Appendix p 12).

## Discussion

This study found that: 55.8% had strong COVID-19 experience and perception and the predictors were being a health worker at a primary/secondary health facility, being a non-clinical health worker, good attitude towards COVID-19 (vaccination), good knowledge about COVID-19, and increase in age; 80.7% had good COVID-19 vaccination expectation and perception and the predictors were being a health worker at a primary/secondary health facility, being a health worker at a public health facility, good attitude towards COVID-19 (vaccination), and good knowledge about COVID-19; and 87.7% had positive COVID-19 vaccination process experience and perception and the predictors were being a health worker at a primary/secondary health facility, being a health worker at a public health facility, having a tertiary education, good attitude towards COVID-19 (vaccination), and good knowledge about COVID-19.

In view of the uniqueness of the above outcome measures explored by the study (the extent and levels of COVID-19 and COVID-19 vaccination and the vaccination process experiences and perceptions), we did not identify relevant studies with similar outcomes for appropriate comparison of findings (descriptive estimates and predictors). However, the moderate level of prevalence (55.8%) of strong COVID-19 experience and perception among the health workers perhaps reflected the fact that the pandemic was relatively less severe in Ebonyi state as fewer COVID-19 cases and related deaths were confirmed compared to many other settings.

The higher prevalence (80.7%) of good COVID-19 vaccination expectation and perception among the health workers despite the misinformation/disinformation and conspiracy theories about COVID-19 and COVID-19 vaccination in the media perhaps indicate that the misinformation/disinformation had limited negative effects on their perception of COVID-19 vaccination. This study was conducted in the context of increased availability and access to actual vaccination. As a result, the real experiences and close perceptions of vaccination attributes (importance, safety/side-effects, and effectiveness) among those who were, or knew others who were, already vaccinated, could have positively affected the health workers' perceptions of the vaccination despite the misinformation/disinformation. However, more studies, particularly qualitative studies, are required to provide more insights in this regard. Moreover, public confidence in the trustworthiness of the social media as a source of COVID-19 information was reported to decline over time during the pandemic [11]. The social media was the primary source of most COVID-19 misinformation/disinformation. Hence, it is plausible to say that the negative effects of COVID-19 and COVID-19 vaccination misinformation/disinformation on health workers' perception of COVID-19 and COVID-19 vaccination decreased over time during the pandemic partly because of decline in their trust in the social media regarding COVID-19. Also, the trust for other health workers who were giving the COVID-19 vaccination and for the government who provided the vaccination were among the five questionnaire items used to measure COVID-19 vaccination expectation and perception. As health workers, the participants would not surprisingly have relatively more trust for the vaccination system (compared to the general public for example) and this would have also enhanced the estimate for the level of COVID-19 vaccination expectation and perception.

Compared to the 80.7% who had good COVID-19 vaccination expectation and perception in our study, only 53.5% of health workers had positive perception of COVID-19 vaccines in another study in Nigeria [12] and 60.5% had good perception of COVID-19 vaccine in a study in Ethiopia [13]. These lower values could have resulted from the different nature, timing, population, and context of the studies compared to our study. The other Nigerian study was institutional based among only health workers in four tertiary hospitals, only online, and conducted much earlier during the initial waves of the pandemic (in late 2020) when there were much confusion and fear and no actual vaccines in Nigeria. Similarly, the Ethiopian study was institutional based, among only health workers in a city (zonal capital), and conducted much earlier (in May 2021) and only about two months after implementation of COVID-19 vaccination in the country. The aforementioned imply that the sociodemographic and professional attributes, and socioeconomic status of the participants in both studies would perhaps be quite different from those in our study. Also, unlike in our study, the participants in the other Nigerian study had no real experiences regarding actual COVID-19 vaccination (in terms of side-effects and effectiveness) while those in the Ethiopian study more likely had very limited real experiences regarding actual COVID-19 vaccination. Moreover, the outcome measure in our study (level of COVID-19 vaccination expectation and perception) with its measurement was quite different from that of the other studies.

The higher prevalence of 87.7% of positive COVID-19 vaccination process experience and perception among the health workers was an encouraging finding and indicate good ease of access and the appeal of the COVID-19 vaccination system to the health workers who are opinion leaders on health matters and are expected to contribute to making the vaccination process more accessible and appealing to the general public. More specifically, 72.7% of the health workers knew a close COVID-19 vaccination place, partly indicating good ease of access. However, in a study in Yemen [14], only 45.4% agreed that they had access to COVID-19 vaccination. This contrasting lower value could be due to the limited availability of COVID-19 vaccination in Yemen perhaps due to conflict-related constraints [14]. Although

we identified no relevant studies to compare with, the above predictors identified by our study indicate factors that should be considered by policy makers and implementers in the strategies to enhance COVID-19 and COVID-19 vaccination and the vaccination process experiences and perceptions among health workers in Ebonyi state, Nigeria and other similar contexts.

In our study, 63.4% were fearful of getting COVID-19, 64.5% said it was possible for them to get COVID-19, 77.3% said it was important for them to receive COVID-19 vaccination, 47.3% were not fearful of having severe side-effects from COVID-19 vaccination, and 72.9% said COVID-19 vaccination would give them protection against COVID-19. Respectively higher proportions were reported by a study in Nigeria [12] in which 91.4% perceived they were at risk of getting COVID-19 and about 80.0% agreed that the vaccine was protective. The contrasting finding could have resulted from the different nature, timing, population, and context of the study as already stated above.

Higher proportions were also reported by foreign studies: about 87.8% were afraid of getting COVID-19 in a study in Ethiopia [15], about 85.0% were afraid of getting COVID-19 at work in China [16], 94.1% thought it was possible for them to get COVID-19 in Saudi [17], 75.7% felt at risk of contracting COVID-19 at work in Saudi [18], and about 87.0% perceived they were at risk of getting COVID-19 in the next one year in the US [19]. Similarly, a higher proportion of 74.5% perceived they were at risk of getting COVID-19 but a slightly lower proportion of 69.9% believed in the effectiveness of the vaccination in another study in Ethiopia [13]. Also, a higher proportion of 89.8% had confidence in the effectiveness of COVID-19 vaccine in Malawi, however, a lower proportion of 41.0% thought it was possible for them to get COVID-19 in the next 12 months (but the study involved only exclusive group of health workers who had been offered the vaccination in their health facilities) [20]. Contrasting lower proportions were reported by a study in Yemen where 69.8% agreed that it was important for them to get vaccinated and 30.9% were not concerned (69.1% were concerned) about the side-effects of the vaccination [14].

The above contrasting findings (by the other previous studies) could perhaps be explained by the different nature, timing, population, and context of these studies as most were only online and involved limited categories of health workers and conducted much earlier, during the initial waves of the pandemic when there was much uncertainty, confusion, fear, and anxiety and when there were limited or no actual COVID-19 vaccines/vaccination. These contexts could have resulted in divergent sociodemographic and professional attributes, socioeconomic status, and real experiences of the study participants regarding COVID-19 and COVID-19 vaccination and differences in perceptions. Most of the studies were conducted when there were no actual COVID-19 vaccination and, as a result, the participants had no real experiences of the importance, safety/side-effects, and effectiveness of COVID-19 vaccination.

Although our study identified some predictors of the above dichotomized positive-negative COVID-19 and COVID-19 vaccination experiences and perceptions, we did not identify any relevant studies for appropriate comparisons as the above relevant and comparable studies did not assess predictors of COVID-19 and COVID-19 vaccination perceptions.

This study had some strengths. This study was not only online but also offline and among the entire populations and categories of health workers, in both rural and urban/semi-urban areas, in Ebonyi state. Hence, the study findings are more generalisable to the general population of health workers in the state and perhaps other poor resource settings with limited internet access. Other strengths were that the outcome measures and the potential covariates were pre-specified in the study protocol which was prospectively registered and prospectively submitted to a peer-review journal before the study was implemented.

One limitation in this study was reporting bias which is associated with questionnaire-based studies. The outcomes were measured by asking participants to report their COVID-19

and COVID-19 vaccination and the vaccination process experiences and perceptions. Hence, they were prone to recall bias because some of these experiences and perceptions were past events. But the bias was minimal because such events were largely recurrent. Also, COVID-19/ COVID-19 vaccination was a controversial topic due to the misinformation/disinformation and conspiracy theories, therefore, there was the tendency for some respondents to exaggerate desirable perceptions and underestimate undesirable perceptions. However, such bias was minimal because the questionnaire was anonymous and the respondents were assured of a high degree of confidentiality. In addition, there was increased possibility of selection bias from the convenience and snowballing sampling and this could limit the generalisability of the study findings to the target population.

## Conclusions

There was moderate level of prevalence of strong COVID-19 experience and perception, high prevalence of good COVID-19 vaccination expectation and perception, and high prevalence of positive COVID-19 vaccination process experience and perception among the health workers during the COVID-19 pandemic in Ebonyi state, Nigeria. The most important predictors of the extent and level of COVID-19 and COVID-19 vaccination and the vaccination process experiences and perceptions were level of primary place of work (primary/secondary versus tertiary facility), level of attitude towards COVID-19 (vaccination), and level of knowledge about COVID-19. Another important predictor was primary place of work (public/private facility). Subsequent COVID-19 and COVID-19 vaccination policy actions in Ebonyi state and Nigeria, and other similar contexts, should be guided by the evidence shown by this study in the strategies to improve health workers' experiences and perceptions of COVID-19 and COVID-19 vaccination and the vaccination process in order to enhance subsequent acceptance/uptake of COVID-19 vaccination and use of other preventive measures. This study's evidence will also guide policy actions/strategies regarding similar diseases in the future.

Subsequent studies on experiences and perceptions of COVID-19 and COVID-19 vaccination, or other infectious diseases/pandemic and their vaccinations, should not only describe prevalence among health workers but should also assess the predictors. Further studies, preferably qualitative studies, are needed on the factors that influence COVID-19 and COVID-19 vaccination (process) experiences and perceptions and particularly on the (positive) effects of health workers' real experiences and close perceptions of COVID-19 vaccination attributes (importance, safety/side-effects, effectiveness) on COVID-19 vaccination expectations and perceptions in the middle of (the negative effects of) misinformation/disinformation and conspiracy theories.

## Supporting information

**S1 Appendix.**
(DOCX)

**S1 Dataset.**
(XLSX)

## Author Contributions

**Conceptualization:** Ugwu I. Omale.

**Data curation:** Ugwu I. Omale.

**Formal analysis:** Ugwu I. Omale.

**Investigation:** Ugwu I. Omale, Cordis O. Ikegwuonu, Ugochi I. A. Nwali, Olaedo O. Nnachi, Okechukwu O. Ukpabi, Ifeyinwa M. Okeke, Richard L. Ewah, Osarhiemen Iyare, Onyinye-chukwu U. Oka, Victor U. Uduma, Azuka S. Adeke.

**Methodology:** Ugwu I. Omale, Cordis O. Ikegwuonu, Glory E. Nkwo, Ugochi I. A. Nwali, Olaedo O. Nnachi, Okechukwu O. Ukpabi, Ifeyinwa M. Okeke, Richard L. Ewah, Osarhie-men Iyare, Chidinma I. Amuzie, Onyinyechukwu U. Oka, Victor U. Uduma, Azuka S. Adeke.

**Project administration:** Ugwu I. Omale.

**Resources:** Ugwu I. Omale.

**Software:** Ugwu I. Omale.

**Validation:** Ugwu I. Omale, Cordis O. Ikegwuonu, Glory E. Nkwo, Ugochi I. A. Nwali, Olaedo O. Nnachi, Okechukwu O. Ukpabi, Ifeyinwa M. Okeke, Richard L. Ewah, Osarhiemen Iyare, Chidinma I. Amuzie, Onyinyechukwu U. Oka, Victor U. Uduma, Azuka S. Adeke.

**Visualization:** Ugwu I. Omale.

**Writing – original draft:** Ugwu I. Omale.

**Writing – review & editing:** Ugwu I. Omale, Cordis O. Ikegwuonu, Glory E. Nkwo, Ugochi I. A. Nwali, Olaedo O. Nnachi, Okechukwu O. Ukpabi, Ifeyinwa M. Okeke, Richard L. Ewah, Osarhiemen Iyare, Chidinma I. Amuzie, Onyinyechukwu U. Oka, Victor U. Uduma, Azuka S. Adeke.

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
