## [Decision Letter · Decision Letter 0]

25 Sep 2023

PONE-D-23-24420COVID-19 and COVID-19 vaccination (process) experiences and perceptions and their predictors among health workers during the COVID-19 pandemic in Ebonyi state, Nigeria: an analytical cross-sectional studyPLOS ONE

Dear Dr. Omale,

Thank you for submitting your manuscript to PLOS ONE. After careful consideration, we feel that it has merit but does not fully meet PLOS ONE’s publication criteria as it currently stands. Therefore, we invite you to submit a revised version of the manuscript that addresses the points raised during the review process.

Abstract

Please summarize the background, it is too robust for an abstract.

From my finding, the KoBoCollect, was accessed using a link through participants WhatsApp and those that didn’t have WhatsApp filled electronic questionnaire. Please, state clearly as such.

Conclusion line 54-58 please, rephrase.

Introduction

Do you mean the reduction in the use and observance of COVID-19 control measures and decrease in the COVID-19 vaccination acceptance by the public?

Line 76-76, please rephrase.

Line 89-92. Please rephrase.

The aim/objective of the study should be the last sentence in the introduction. Please, clearly state it in a sentence.

Delete line 108- “and the report of the findings are presented in this paper”

Methods

Study design and participants

Line 112- “all categories of health workers” please, state the categories of health workers interviewed clearly.

Clinical and nonclinical staff in the private health sector should be another sentence.

Line 114-“Eligible health workers”, inclusion criteria? Clearly state your inclusion and exclusion criteria for the study.

Line 114-115 “Eligible health workers were those who work or live in Ebonyi state and gave verbal consent and they were selected by convenience and snowballing techniques”

Selection process; ”sampling technique” applied for the study should be separated from eligibility criteria.

Line 114-115 “The study protocol is described elsewhere.7” state the link or cite the protocol that was used.

Line 115-123 should be under the topic data collection

Line 124 Independent factors and outcome measures; this should be under the title questionnaire as details that follow described the various sections of the questionnaire while the rest of the explanation should be part of the data analysis as it clearly described how the data from the responses were graded after analysis.

I noticed that there are a lot of repetitions with regard to the questionnaire

Clearly define your methodology:

• Study site/ study area: some details on the study site and health facilities in Ebonyi State.

• Study design

• Inclusion and exclusion criteria

• Sampling techniques

• Questionnaire design

• Ethical clearance and consent for participation

• Data management

o Data collection

o Data analysis

Results

Line 276, 282- please, do not start a sentence with a figure

Discussion

Line 295-297 seems like the aim of the study, please delete to avoid repetition.

Line 311-314, please refer the outcome of the study to Ebonyi State rather than generalizing it to Nigeria since the study was conducted in Ebonyi state and the method of sampling was not randomized.

Line 315-319 could it be generally classified as infodemic with regard to COVI-19 and COVID-19 vaccination?

Limitations of the study should be the last paragraph of the discussion.

Discuss the strength of the study before the limitations of the study.

Line 418-420 “Hence, the study findings are more generalizable to the general population of health workers in the state, other states in Nigeria, and other poor resource settings with limited internet access.”

The above statement mentioned that this study could be generalized to other states in Nigeria. Going by the sampling method that was used in the study could we generalize it to Nigeria? It will be more appropriate to limit our findings to Ebonyi State.

Conclusion

Line 438-446, this should be stated as part of the limitation of the study and also the appeal for further studies to cover the qualitative aspect of the current study. Rephrase and make it more concise to fit into part of the limitation of the study.

We look forward to receiving your revised manuscript.

Kind regards,

Ayi Vandi Kwaghe, D.V.M., M.V.Sc., P.G.D.E. Ph.D., MPH

Academic Editor

PLOS ONE

Journal Requirements:

Reviewers' comments:

Reviewer's Responses to Questions

**Comments to the Author**

1. Is the manuscript technically sound, and do the data support the conclusions?

Reviewer #1: Partly

Reviewer #2: No

2. Has the statistical analysis been performed appropriately and rigorously? 

Reviewer #1: Yes

Reviewer #2: I Don't Know

3. Have the authors made all data underlying the findings in their manuscript fully available?

Reviewer #1: Yes

Reviewer #2: No

4. Is the manuscript presented in an intelligible fashion and written in standard English?

Reviewer #1: No

Reviewer #2: Yes

5. Review Comments to the Author

Reviewer #1: check your reference

Discussion:

- Please, discuss your results in details.

- Mention the limitations in study

Conclusion

Please rewrite conclusion again to be more informative.

Overall:

- I would advise that the manuscript is checked for grammatical and spelling errors

Reviewer #2: I can see that a lot of work has gone into this paper, the topic is relevant and pertinent. Unfortunately my sense is that at the moment it is not sufficiently rigorously enough designed, i.e. in terms of the meanings of the responses to illustrate new knowledge. The authors may wish to use some of the clearer questions to reanalyse the data to enhance rigour.

Title

Descriptive, appropriate, if a little long, could a few ands be omitted ?

Abstract

It’s detailed, but could benefit from a little clarification and a few more commas in places, or shorter sentences, as per sections highlighted in yellow.

‘Background: The COVID-19 pandemic continues to be a global emergency with

millions of new infections/re-infections and thousands of related deaths occurring

weekly globally and requires long-term management/control. One of the factors

sustaining the pandemic is the reduction in the use/non-use of control measures.

Health workers continue to be at high risk of contracting COVID-19 because of their

key roles in the management/control of the new COVID-19 infections/re-infections’

State needs capital letter, needs breaking up into shorter sentences.

‘Methods: We conducted an analytical online-offline cross-sectional survey between

March 12 and May 9, 2022 among all categories of health workers (clinical/non-clinical,

public/private) working/living in Ebonyi state who consented to participate and were

selected by convenience/snowballing techniques. A structured self-administered and

interviewer-administered electronic questionnaire, respectively via WhatsApp and

KoBoCollect, was used to collect data which was analysed using descriptive statistics

and bivariate/multivariate generalized linear models’

‘strong COVID-19 experience and perception, 80.7% had good COVID-19 vaccination expectation and

perception, and 87.7% had positive COVID-19 vaccination process experience and

perception. The most important predictors of the extent/level of COVID-19 and COVID19 vaccination (process) experiences and perceptions were level of place of work

(primary-secondary/tertiary), level of attitude towards COVID-19 (vaccination), and

level of knowledge about COVID-19. Another important predictor was place of work

(public/private).

Conclusions: There was moderate level of prevalence of strong COVID-19 experience

and perception, high prevalence of good COVID-19 vaccination expectation and

perception, and high prevalence of positive COVID-19 vaccination process experience

and perception among the health workers during the COVID-19 pandemic in Ebonyi

state, Nigeria’.

It might be advisable to reduce the contents of the abstract, as its not clear what some of these results mean in practice, e.g. ‘strong C-19 experience and perception‘

Intro

This is repetitious, and could be shortened e.g.

‘An understanding of health workers’ COVID-19 experiences and perceptions, and the determinants, would be useful in the subsequent planning of tailored COVID-19 behaviour change communication strategies’

Could the authors say what the determinants refers to, also what behaviour change they are referring to- is it increased vaccine take-up

Methods

Unfortunately the categories for many of the resulting questions are too vague to add new knowledge, what does ‘close’ ‘good’ ‘poor’ mean.

Results

As above

Discussion

Conclusion

At the moment this paper is not ready for publication. I'm not an expert on survey’s so perhaps I'm not the person to be reviewing here. The number of responses is very impressive, and the time and energy spent on recruitment and data analysis is impressive. However, my sense is that the questions and responses presented here are too vague to illustrate new knowledge.

If the authors can utilise and analyse some survey questions with really detailed, specific questions, and resubmit, with a clear aim, that could illustrate new knowledge, my sense is at the moment its not there.

6. PLOS authors have the option to publish the peer review history of their article (what does this mean?). If published, this will include your full peer review and any attached files.

Reviewer #1: No

Reviewer #2: No

---

## [Author Response · Author response to Decision Letter 0]

6 Nov 2023

REVIEWERS’ COMMENTS AND RESEARCHERS’ RESPONSES 

Thank you for your comments and for creating the time to review our manuscript. 

Academic Editor

Abstract

1. Comment: Please summarize the background, it is too robust for an abstract. 

Response: The background has been reduced

2. Comment: From my finding, the KoBoCollect, was accessed using a link through participants WhatsApp and those that didn’t have WhatsApp filled electronic questionnaire. Please, state clearly as such.

Response: It has been modified as appropriate (an electronic questionnaire administered via WhatsApp or KoboCollect for those that did not have WhatsApp)

3. Comment: Conclusion line 54-58 please, rephrase. 

Response: It has been rephrased

Introduction

1. Comment: Do you mean the reduction in the use and observance of COVID-19 control measures and decrease in the COVID-19 vaccination acceptance by the public?

Line 76-76, please rephrase.

Line 89-92. Please rephrase. 

Response: Yes. The statement regarding use of control measures refers to the general population which include the health workers. Both have been modified as appropriate.

2. Comment: The aim/objective of the study should be the last sentence in the introduction. Please, clearly state it in a sentence. Delete line 108- “and the report of the findings are presented in this paper” 

Response: The phrase has been deleted and the statements regarding the aim has been modified as suggested. 

Methods

Study design and participants

1. Comment: Line 112- “all categories of health workers” please, state the categories of health workers interviewed clearly. Clinical and nonclinical staff in the private health sector should be another sentence. 

Response: We understand or believe the statement “… all categories of health workers, both clinical and non-clinical staff in public and private health care sectors, ...” is appropriately stated. The phrases “both clinical and non-clinical” and “public and private” already define the preceding phrase “all categories”. The clinical and non-clinical staff are respectively listed in the legend of table 1, including their categorization into public and private. Should the health workers still be listed in this section as they are in the legend of table 1? 

2. Comment: Line 114-“Eligible health workers”, inclusion criteria? Clearly state your inclusion and exclusion criteria for the study.

Response: The eligibility criteria (which is also called inclusion and exclusion criteria) are already clearly stated. Please note that eligibility criteria can either be stated as “eligibility criteria” or separated into “inclusion and exclusion criteria”. We chose the former, and used it in the published protocol, as the latter is not always necessary and do not always add value (Reference: STROBE reporting guidelines). It will not be appropriate to separate it into the “inclusion and exclusion criteria” format at this stage.

3. Comment: Line 114-115 “Eligible health workers were those who work or live in Ebonyi state and gave verbal consent and they were selected by convenience and snowballing techniques”

Selection process; ”sampling technique” applied for the study should be separated from eligibility criteria. 

Response: We used the unstructured format to reduce the number of subheadings and word count. They have been separated into different sentences in line with the suggestion.

4. Comment: Line 114-115 “The study protocol is described elsewhere.7” state the link or cite the protocol that was used.

Response: We seem not to understand this comment because the protocol is already cited (citation 7).

5. Comment: Line 115-123 should be under the topic data collection 

Response: The statements in lines 112–119 (not 115–123) of the original manuscript were meant to describe the convenience and snowballing sampling technique. Some of them also fit into the “data collection” subsection. They have been merged under one subheading in line with other subsequent comments.

6. Comment: Line 124 Independent factors and outcome measures; this should be under the title questionnaire as details that follow described the various sections of the questionnaire while the rest of the explanation should be part of the data analysis as it clearly described how the data from the responses were graded after analysis.

I noticed that there are a lot of repetitions with regard to the questionnaire

Response: Please note that it will not be appropriate at all to use “questionnaire” as a subheading here because this section did not describe sections of the questionnaire. It rather described the relevant independent and outcome factors (regarding the aim of this study) which were created from items in the questionnaire and the explanations were descriptions of how these factors were created. For example, level of knowledge and level of attitude (as independent factors) and extent of COVID-19 experience and perception, level of COVID-19 vaccination expectation and perception, and level of COVID-19 vaccination process experience and perception (as outcome factors) were not questionnaire items. Also, the creation of these variables was done before analysis and ideally falls under “data management” and not “statistical analyses” and because of the extent of these variable creation, we believe it is better to put it under a separate subheading than “statistical analyses”. The methods section has been rearranged in response to other subsequent comments.

We do not understand what is meant by “a lot of repetitions with regard to the questionnaire” because there are no such repetitions. Please not that this section is not (and should not be seen as) description of sections of the questionnaire (which was under data collection). It is rather a description of the relevant independent and outcome factors which were created from items in the questionnaire. Also, extent/level of COVID-19 and COVID-19 vaccination (process) experiences and perceptions were a different set of outcome measures from the dichotomized positive versus non-positive categories of COVID-19 and COVID-19 vaccination (process) experiences and perceptions.

7. Comment: Clearly define your methodology:

• Study site/ study area: some details on the study site and health facilities in Ebonyi State.

• Study design

• Inclusion and exclusion criteria

• Sampling techniques

• Questionnaire design

• Ethical clearance and consent for participation

• Data management

o Data collection

o Data analysis

Response: The structure/subheadings in the method section has been modified as suggested. However, kindly note that the modification is also: in consideration of PLOS ONE Guidelines to authors to limit manuscript sections and sub-sections to 3 heading levels, in line with our foregoing responses, and in consideration of the fact that this study was nested in a larger study with a published protocol (and the need to maintain reasonable fidelity to the protocol). We do not understand what is actually meant by “questionnaire design”. The relevant sections of the questionnaire are already stated under data collection.

Results

Comment: Line 276, 282- please, do not start a sentence with a figure 

Response: There were not new sentences but part of the statements under the “colon” in the opening sentence but the “semi-colons” to indicate these were omitted. These have been corrected.

Discussion

1. Comment: Line 295-297 seems like the aim of the study, please delete to avoid repetition.

Response: It has been deleted.

2. Comment: Line 311-314, please refer the outcome of the study to Ebonyi State rather than generalizing it to Nigeria since the study was conducted in Ebonyi state and the method of sampling was not randomized.

Response: It has been modified as suggested.

3. Comment: Line 315-319 could it be generally classified as infodemic with regard to COVI-19 and COVID-19 vaccination? 

Response: Misinformation simply refers to false information. Disinformation is a specific type of misinformation created with the intention to deceive (that is, deliberate spread of false information). Both are preferred in this context. Infodemic simply means too much of information including false information and would not properly capture what we intend to emphasize.

4. Comment: Limitations of the study should be the last paragraph of the discussion.

Discuss the strength of the study before the limitations of the study.

Response: The order of the limitation and strength has been changed as suggested.

5. Comment: Line 418-420 “Hence, the study findings are more generalizable to the general population of health workers in the state, other states in Nigeria, and other poor resource settings with limited internet access.”

The above statement mentioned that this study could be generalized to other states in Nigeria. Going by the sampling method that was used in the study could we generalize it to Nigeria? It will be more appropriate to limit our findings to Ebonyi State.

Response: The generalization as used in this context was in comparison to the only online studies as our study was both online and offline and thus, more representative of the general population of health workers (in urban and rural areas). However, it has been modified.

Conclusion

Comment: Line 438-446, this should be stated as part of the limitation of the study and also the appeal for further studies to cover the qualitative aspect of the current study. Rephrase and make it more concise to fit into part of the limitation of the study. 

Response: We do not see this as limitation that should be placed under the “limitation” section. Such recommendation for further studies are common in relevant literature and are based on all the available evidence (from our study and other previous studies) as discussed under the “discussion” section.

Reviewer # 1 

Comment: check your reference 

Response: We do not understand what specific issue to check in the reference because no specific comment or suggestion was made. However, please note that we used software to manage the referencing and will crosscheck as appropriate.

Discussion

1. Comment: Please, discuss your results in details. 

Response: We strongly believe the results have already been appropriately discussed in details.

2. Comment: Mention the limitations in study

Response: The limitations are already mentioned. Please see lines 413–423.

Conclusion

Comment: Please rewrite conclusion again to be more informative. 

Response: We do not understand the comment. The conclusion is based on the study findings and the discussion of all available evidence.

Overall

Comment: I would advise that the manuscript is checked for grammatical and spelling errors 

Response: The manuscript has been proof read again for typos and other errors 

Reviewer # 2 

Comment: I can see that a lot of work has gone into this paper, the topic is relevant and pertinent. Unfortunately my sense is that at the moment it is not sufficiently rigorously enough designed, i.e. in terms of the meanings of the responses to illustrate new knowledge. The authors may wish to use some of the clearer questions to reanalyse the data to enhance rigour.

Response: Please note that, as stated in the last part of the introduction section, this study was nested in a much larger study which was pre-registered before implementation and has a published study protocol which is also cited in this manuscript. As also stated in the “study strength” section, the outcomes and potential covariates were pre-specified in the study protocol. The statement “The authors may wish to use some of the clearer questions to reanalyse the data to enhance rigour” perhaps was due to lack of basic understanding of our study concepts and analyses. 

Regarding the comment about new knowledge, please note the following:

(a) “New knowledge” is a spectrum and that in most cases, especially in applied disciplines/research, research evidence are not completely/absolutely new 

(b) In many cases research evidence generation is incremental with subtle additions

(c) In most cases the assessment of “new knowledge” or “novelty” is context-specific 

(d) Research evidence in many (perhaps in most) cases are corroborative of existing evidence and need not to be completely/absolutely new to be of scientific or public health or health policy importance as the importance of any evidence is also context-specific. 

Title

Comment: Descriptive, appropriate, if a little long, could a few ands be omitted?

Response: The study is analytical, not descriptive. Please note that there should be no inferential analysis (use of test statistics) in descriptive studies which should only be “descriptive” as the name indicate.

The title is not longer than necessary considering the nature/purpose of the study. Rather, in line with STROBE guidelines, the title appropriately captures the study design and research questions in addition to the participants and study setting. 

Abstract

Comment: It’s detailed, but could benefit from a little clarification and a few more commas in places, or shorter sentences, as per sections highlighted in yellow. 

‘Background: The COVID-19 pandemic continues to be a global emergency with

millions of new infections/re-infections and thousands of related deaths occurring

weekly globally and requires long-term management/control. One of the factors

sustaining the pandemic is the reduction in the use/non-use of control measures.

Health workers continue to be at high risk of contracting COVID-19 because of their

key roles in the management/control of the new COVID-19 infections/re-infections’

State needs capital letter, needs breaking up into shorter sentences.

Response: Shorter sentences cannot enhance clarification because the variable names are long. There is no highlighted section in the pasted quote. However, the abstract has been modified in line with the suggestions of the academic editor.

Please note that the primary word is “state” but “State” can also be used, when referring to a country (e.g Nigeria) or part of a country (e.g Ebonyi). Writing it as “Ebonyi state” is very okay. This can easily be confirmed using standard dictionaries e.g Oxford learners dictionaries. From experience, we very much understand that many scholars, particularly in Nigeria, do not understand the usage of the word as we have explained above.

Methods, Results, Conclusion

1. Comment: It might be advisable to reduce the contents of the abstract, as its not clear what some of these results mean in practice, e.g. ‘strong C-19 experience and perception‘ 

Response: There are no highlighted sections in all the pasted quotes. However, the abstract has been modified in line with the suggestions of the academic editor.

Also note that, as stated in the manuscript text under the methods section, the related questionnaire items were scored and the scores were summed and categorized (dichotomized) as strong versus not strong COVID-19 experience and perception using the 50th percentile as cutoff. 

Introduction

Comment: This is repetitious, and could be shortened e.g. ‘An understanding of health workers’ COVID-19 experiences and perceptions, and the determinants, would be useful in the subsequent planning of tailored COVID-19 behaviour change communication strategies’ Could the authors say what the determinants refers to, also what behaviour change they are referring to- is it increased vaccine take-up

Response: Please note that due to the nature of the study and the concepts that were explored, the manuscript and manuscript sections are better understood as a whole. For example, to understand the statements in the above comment, you have to understand (and remember) all the preceding statements in the introduction. 

For example, the preceding statement in lines 81–84 (of the revised manuscript) contains the behaviours being referred to subsequently in lines 97–98 and to continue to restate them will unnecessarily increase word count.

Also note that the determinants being referred to where the potential determinants the study set out to identify which could not be stated here under the introduction but rather presented as part of the findings in the result section.

Methods, Results

Comment: Unfortunately the categories for many of the resul

---

## [Decision Letter · Decision Letter 1]

1 Mar 2024

PONE-D-23-24420R1COVID-19 and COVID-19 vaccination (process) experiences and perceptions and their predictors among health workers during the COVID-19 pandemic in Ebonyi state, Nigeria: an analytical cross-sectional studyPLOS ONE

Dear Dr. Omale,

Thank you for submitting your manuscript to PLOS ONE. After careful consideration, we feel that it has merit but does not fully meet PLOS ONE’s publication criteria as it currently stands. Therefore, we invite you to submit a revised version of the manuscript that addresses the points raised during the review process.

**ACADEMIC EDITOR: **

Study design and participants

“All categories of health workers” please, state the categories of health workers interviewed clearly.

It’s okay to have it in Table 1 of your results but you need to categorically state this in your methods for researchers that will be reading your article. Please clearly state these categories in your methods.

“Eligible health workers”, inclusion criteria? Clearly state your inclusion and exclusion criteria for the study.

Does this mean that all health workers in those facilities that gave their verbal consent were eligible without taken into consideration of other factors that may affect the parameters such as confounders? Were there people that met the inclusion criteria but could not be included in the study due to some reasons? Exclusion criteria is relevant to a study and I strongly suggest that is included in this manuscript.

The study protocol has been published, please provide the link if any. If there is no link to directly download the protocol, the citation will do.

We look forward to receiving your revised manuscript.

Kind regards,

Ayi Vandi Kwaghe, D.V.M., M.V.Sc., P.G.D.E. Ph.D., MPH, FETP

Academic Editor

PLOS ONE

Journal Requirements:

Additional Editor Comments:

Study design and participants

“All categories of health workers” please, state the categories of health workers interviewed clearly.

It’s okay to have it in Table 1 of your results but you need to categorically state this in your methods for researchers that will be reading your article. Please clearly state these categories in your methods.

“Eligible health workers”, inclusion criteria? Clearly state your inclusion and exclusion criteria for the study.

Does this mean that all health workers in those facilities that gave their verbal consent were eligible without taken into consideration of other factors that may affect the parameters such as confounders? Were there people that met the inclusion criteria but could not be included in the study due to some reasons? Exclusion criteria is relevant to a study and I strongly suggest that is included in this manuscript.

The study protocol has been published, please provide the link if any. If there is no link to directly download the protocol, the citation will do.

Reviewers' comments:

Reviewer's Responses to Questions

**Comments to the Author**

1. If the authors have adequately addressed your comments raised in a previous round of review and you feel that this manuscript is now acceptable for publication, you may indicate that here to bypass the “Comments to the Author” section, enter your conflict of interest statement in the “Confidential to Editor” section, and submit your "Accept" recommendation.

Reviewer #3: All comments have been addressed

Reviewer #4: (No Response)

2. Is the manuscript technically sound, and do the data support the conclusions?

Reviewer #3: Partly

Reviewer #4: Yes

3. Has the statistical analysis been performed appropriately and rigorously? 

Reviewer #3: Yes

Reviewer #4: I Don't Know

4. Have the authors made all data underlying the findings in their manuscript fully available?

Reviewer #3: No

Reviewer #4: Yes

5. Is the manuscript presented in an intelligible fashion and written in standard English?

Reviewer #3: No

Reviewer #4: Yes

6. Review Comments to the Author

Reviewer #3: - Introduction is long

- Results need to be presented in more simplified and concise way

- Bias with questionnaire, No objective criteria

-

Reviewer #4: (No Response)

7. PLOS authors have the option to publish the peer review history of their article (what does this mean?). If published, this will include your full peer review and any attached files.

Reviewer #3: **Yes: **Mohamed Saad Hashim

Reviewer #4: **Yes: **Mohammed Isa Bammami

---

## [Author Response · Author response to Decision Letter 1]

6 Mar 2024

RESPONSES TO COMMENTS BY REVIEWERS 

Thank you for creating the time to review our manuscript and for your comments which have contributed to improve the manuscript. 

Academic Editor 

Study design and participants

1. Comment: “All categories of health workers” please, state the categories of health workers interviewed clearly. It’s okay to have it in Table 1 of your results but you need to categorically state this in your methods for researchers that will be reading your article. Please clearly state these categories in your methods.

Response: The individual categories of health workers have been stated in the methods section.

2. Comment: “Eligible health workers”, inclusion criteria? Clearly state your inclusion and exclusion criteria for the study. Does this mean that all health workers in those facilities that gave their verbal consent were eligible without taken into consideration of other factors that may affect the parameters such as confounders? Were there people that met the inclusion criteria but could not be included in the study due to some reasons? Exclusion criteria is relevant to a study and I strongly suggest that is included in this manuscript.

Response: Just as we said before, “eligibility criteria” (not “inclusion and exclusion criteria”) was used in the study design and during study implementation. Thus, it will not be appropriate to create “inclusion and exclusion criteria” at this stage. Such practice was informed by the STROBE reporting guidelines, relevant literature and our understanding that eligibility criteria could either be stated as “eligibility criteria” or separated into “inclusion and exclusion criteria”. 

Please note: 

(i) Potential confounders were already factored into the eligibility criteria (e.g living or working in Ebonyi state)

(ii) Because we used the “eligibility criteria” and not the “inclusion and exclusion criteria”, there was no separate inclusion criteria to be met, and exclusion criteria for excluding those who were included. All those living or working in Ebonyi state who gave consent were eligible.

(iii) If we were to use the “inclusion and exclusion criteria” format, the study population would perhaps be stated differently. Example:

(a) Study population: the health workers in Ebonyi state

(b) Inclusion criteria: clinical and non-clinical staff in public and private health care sectors such as: ……… (the individual categories of health workers could be listed here)

(c) Exclusion criteria: Not working or living in Ebonyi state, withholding of consent.

(iv) There are also examples of the “eligibility criteria” format in the literature (e.g https://journals.plos.org/plosone/article?id=10.1371/journal.pone.0275234)

(v) The “inclusion and exclusion criteria” format can easily be restated or modified to the “eligibility criteria” format. Example: The “inclusion and exclusion criteria” is this paper (https://journals.plos.org/plosone/article?id=10.1371/journal.pone.0289825) is shown below:

Inclusion criteria:

“In order to participate in the study, participants must be at least 18 years old, have type 2 diabetes, be getting outpatient treatment, have fasting blood sugar (FBG) and glycated hemoglobin (HbA1C) tests, be taking diabetes medication, and have obtained consent.”

Exclusion criteria:

“Pregnant women and those who experienced an emergency requiring hospitalization are excluded.”

The above “inclusion and exclusion criteria” could be stated as “eligibility criteria” as below:

In order to participate in the study, participants must be at least 18 years old, have type 2 diabetes, be getting outpatient treatment, have fasting blood sugar (FBG) and glycated hemoglobin (HbA1C) tests, be taking diabetes medication, not be pregnant, not experienced an emergency requiring hospitalization, and have obtained consent.”

(vi) “Eligibility criteria” (not “inclusion and exclusion criteria”) was included in the STROBE Checklist and in the explanation and elaboration it was noted that eligibility criteria could also be presented as “inclusion and exclusion criteria” even though this separation was not always necessary or useful (https://journals.plos.org/plosmedicine/article?id=10.1371/journal.pmed.0040297)

Reviewer

1. Comment: Title Clarity: The title is quite long and complex. Consider simplifying it for clarity while still capturing the essence of the study. For instance, "Experiences and Perceptions of COVID-19 and Vaccination Among Health Workers in Ebonyi State, Nigeria: A Cross-Sectional Analysis." Avoid Redundancies: The title repeats "COVID-19" twice, which could be streamlined for brevity.

Response: The title has been modified to reduce the length as we deem appropriate, to maintain consistency with our study concepts. Please note that the study explored experiences and perceptions/expectations about three main factors: COVID-19, COVID-19 vaccination (or vaccine), and COVID-19 vaccination process. These three factors were distinctly assessed and presented in the paper. The title was initially framed to capture these factors.

The suggested title is not acceptable to us because it does not clearly reflect our study concepts. “COVID-19” and “COVID-19 vaccination” are distinct and have different meaning and both have to be captured by the title. There is no recognized research guidelines/recommendations that a word should only appear once in a research title and examples of titles with a word appearing multiple (two or more) times is common in the scientific literature. What is more important is the overall meaning of the title taken as a whole (read and understood as a whole) and not the number of times a word appear.

2. Comment: It would be nice if you highlighted the implications of your findings in the abstract.

Response: The implications are the conclusions which are already stated. However, the conclusions have been modified as appropriate.

3. Comment: Some sentences are lengthy and could be made more concise. Shorter, clearer sentences would improve readability.

4. Comment: Page 3, Paragraph 1, Lines 60-61: The sentence beginning "Despite the successes recorded..." is quite long and complex. It could be broken down into shorter sentences for better readability.

Response: The first paragraph has been revised in line with new literature and some sentences have been reframed or broken into two to reduce their length. The main concepts investigated were variables with long names and this contributed to the length of sentences.

5. Comment: Page 3, Paragraph 2, Lines 72-73: The phrase "One of the factors sustaining..." is a bit vague. It might be helpful to specify which particular control measures are being referred to, and how they directly relate to the pandemic's management.

Response: The statement was based on the information in the cited reference material and it was modified to the current form in line with the comment of the Academic Editor during the first revision. We understand that it is clear as stated.

6. Comment: Limitations of Methodology: While some limitations are inherent in the study design (like sampling method), it would be helpful to discuss these limitations more explicitly. This includes addressing the potential biases and constraints that might affect the interpretation of the findings.

7. Comment: Study Design and Participants: The study is described as an analytical cross-sectional survey among health workers in Ebonyi State. The inclusion of both clinical and non-clinical staff is commendable for a comprehensive understanding. However, the use of convenience and snowballing sampling techniques might limit the generalizability of the findings. It’s important to discuss how this sampling approach may influence the results.

Response: We have already stated that selection bias from the sampling approach was a limitation (in addition to the other limitations) under “limitation” in the discussion section. We did not think there was much to discuss regarding the selection bias or that it was necessary to do so. The implication of the selection bias is obvious and well known. However, the statement has been modified and the implication added.

8. Comment: Data Collection Process: The use of a structured electronic questionnaire, distributed via WhatsApp and KoBoCollect, is innovative and suitable for the current digital age. However, there might be a need to elaborate on how the questionnaire was developed and validated to ensure it accurately captures the required information.

Response: Some statements have been added regarding questionnaire design and validation (lines 136–139) but the details are in the cited study protocol.

9. Comment: Statistical Analysis: The manuscript mentions the use of descriptive statistics, bivariate, and multivariate generalized linear models for data analysis. It would be beneficial to provide more details about the variables included in the models and the rationale for choosing these specific statistical methods.

Response: It is stated that all the independent factors were added to the model in the adjusted analysis. Please note that these independent factors are already listed in a preceding paragraph (the first paragraph under the “data management and statistical analyses” section. It will be an unnecessary repetition to list these factors again when talking about adjusted analyses (because all of them were inputed in the model as already stated).

10. Comment: Limitations of Data: Acknowledge any limitations inherent in your data, such as those arising from the sampling method or response biases.

Response: Identified limitations have already been stated in the discussion section.

11. Comment: Discussion of Key Findings: Briefly interpret the significance of these findings in the context of your research objectives.

Response: We understand that this has already been done.

---

## [Decision Letter · Decision Letter 2]

22 Apr 2024

COVID-19 and COVID-19 vaccination experiences and perceptions among health workers during the pandemic in Ebonyi state, Nigeria: an analytical cross-sectional study

PONE-D-23-24420R2

Dear Dr. Omale,

We’re pleased to inform you that your manuscript has been judged scientifically suitable for publication and will be formally accepted for publication once it meets all outstanding technical requirements.

Kind regards,

Ayi Vandi Kwaghe, D.V.M., M.V.Sc., P.G.D.E. Ph.D., MPH, FETP

Academic Editor

PLOS ONE

Additional Editor Comments (optional):

Reviewers' comments:

Reviewer's Responses to Questions

**Comments to the Author**

1. If the authors have adequately addressed your comments raised in a previous round of review and you feel that this manuscript is now acceptable for publication, you may indicate that here to bypass the “Comments to the Author” section, enter your conflict of interest statement in the “Confidential to Editor” section, and submit your "Accept" recommendation.

Reviewer #3: (No Response)

Reviewer #4: All comments have been addressed

2. Is the manuscript technically sound, and do the data support the conclusions?

Reviewer #3: Yes

Reviewer #4: Yes

3. Has the statistical analysis been performed appropriately and rigorously? 

Reviewer #3: Yes

Reviewer #4: I Don't Know

4. Have the authors made all data underlying the findings in their manuscript fully available?

Reviewer #3: Yes

Reviewer #4: Yes

5. Is the manuscript presented in an intelligible fashion and written in standard English?

Reviewer #3: Yes

Reviewer #4: Yes

6. Review Comments to the Author

Reviewer #3: The discussion is long. It needs to be considered and targeted. The tables need to be more simplified.

Reviewer #4: (No Response)

7. PLOS authors have the option to publish the peer review history of their article (what does this mean?). If published, this will include your full peer review and any attached files.

Reviewer #3: **Yes: **Mohamed Saad Hashim

Reviewer #4: **Yes: **Mohammed Isa Bammami

---

## [Editor Report · Acceptance letter]

26 Apr 2024

PONE-D-23-24420R2 

PLOS ONE

Dear Dr. Omale, 

I'm pleased to inform you that your manuscript has been deemed suitable for publication in PLOS ONE. Congratulations! Your manuscript is now being handed over to our production team.

Kind regards, 

on behalf of

Dr. Ayi Vandi Kwaghe 

Academic Editor

PLOS ONE